# Structure-based rational design of an enhanced fluorogen-activating protein for fluorogens based on GFP chromophore

Marina V. Goncharuk[1,5], Nadezhda S. Baleeva[1,5], Dmitry E. Nolde [1,2], Alexey S. Gavrikov[1], Alexey V. Mishin [3], Alexander S. Mishin [1], Andrey Y. Sosorev [1], Alexander S. Arseniev[1], Sergey A. Goncharuk [1,3], Valentin I. Borshchevskiy [3], Roman G. Efremov[1,2,3], Konstantin S. Mineev [1,3,6✉] & Mikhail S. Baranov [1,4,6✉]

"Fluorescence-Activating and absorption-Shifting Tag" (FAST) is a well-studied fluorogen-activating protein with high brightness and low size, able to activate a wide range of fluorogens. This makes FAST a promising target for both protein and fluorogen optimization. Here, we describe the structure-based rational design of the enhanced FAST mutants, optimized for the **N871b** fluorogen. Using the spatial structure of the FAST/**N871b** complex, NMR relaxation analysis, and computer simulations, we identify the mobile regions in the complex and suggest mutations that could stabilize both the protein and the ligand. Two of our mutants appear brighter than the wild-type FAST, and these mutants provide up to 35% enhancement for several other fluorogens of similar structure, both in vitro and in vivo. Analysis of the mutants by NMR reveals that brighter mutants demonstrate the highest stability and lowest length of intermolecular H-bonds. Computer simulations provide the structural basis for such stabilization.

[1] Shemyakin-Ovchinnikov Institute of Bioorganic Chemistry RAS, Moscow 117997, Russia. [2] National Research University Higher School of Economics, Moscow 101000, Russia. [3] Moscow Institute of Physics and Technology, Dolgoprudny 141701, Russia. [4] Pirogov Russian National Research Medical University, Moscow 117997, Russia. [5] These authors contributed equally: Marina V. Goncharuk and Nadezhda S. Baleeva. [6] These authors jointly supervised this work: Konstantin S. Mineev and Mikhail S. Baranov. ✉email: mineev@nmr.ru; baranovmikes@gmail.com

Fluorogen-activating proteins (FAPs) represent an important class of instruments in modern molecular biology. Unlike the fluorescent proteins (FPs), they do not contain a chromophore and become active only upon interaction with special low-molecular-weight compounds, fluorogens. Fluorogens themselves are usually dim and become bright only after their binding to FAP. In contrast to FPs, FAPs do not need time and oxygen to mature and are in general smaller, which decreases their chance to distort the behavior of the object under investigation. Thus, FAP is a very convenient genetically encoded fluorescent label that can be switched on and off on demand. Moreover, while the performance of a given FP may be altered only via the protein mutagenesis, the optic properties of a FAP may be additionally changed by the variation of the fluorogen structure, which broadens substantially the possible range of emission and absorption maxima.

Out of all FAPs, the so-called "Fluorescence-Activating and absorption-Shifting Tag" (FAST) received a lot of attention during the past several years[1–3]. The protein complexes with fluorogens are relatively bright, comparable to the brightness of some FPs[4–6]. Besides, it can bind a vast class of chromophores and has a rather small size of ~14 kDa, which makes it a promising target for both protein and fluorogen optimization. In particular, dozens of FAST ligands with various absorption/emission properties have been proposed[7–11], several FAST mutants, optimized for different ligands have been found[10–14] and a split-construct of FAST was designed[15]. The latest studies report the FAST mutants, redFAST, and green-FAST, applicable for the orthogonal fluorescent labeling[13], and a promiscuous variant of FAST, pFAST, capable of efficient binding of several ligands of different color[10].

Almost all of the FAST improvements made to date, have been achieved via the random mutagenesis/directed evolution approach. This technique is highly efficient since it allows covering a vast variety of mutants in a single experiment, and it is really obvious, looking at the results of the recent works in this area[10]. On the other hand, the techniques used for the selection of the enhanced clones are not ideal, and some favorable mutations may be lost in experiments of such kind. Furthermore, in most cases, the mutagenesis leads to a protein variant that is optimal for a particular fluorogen or a group of compounds, and has poor performance with other possible ligands[13]. Random mutagenesis of FAST for each out of dozens of ligands may be too time-consuming. The structure-driven rational design may be considered as an alternative to the random mutagenesis, however, this pathway was hindered for FAST until recently, when our group published the first spatial structure of the protein in the apostate and in complex with the ligand N871b[16]. Using the spatial structure, we managed to design the shorter variant of FAST, nanoFAST, and find an appropriate fluorogen. Here, we utilize the spatial structure, NMR relaxation analysis, and molecular dynamics simulations to rationally design an improved variant of FAST, optimized for the N871b fluorogen.

## Results

### Identification of "hot spots" in FAST/N871b complex.
For rational mutagenesis, one needs first to identify the "hot spots" in the protein structure. It is well known that the brightness of fluorophores depends on the extinction coefficient and the quantum yield of fluorescence. In contrast, the brightness of FAPs typically just slightly depends on the dissociation constant[10–13], therefore it is hard to predict the effect of mutations, considering only the free energy of protein/ligand interactions. On the other hand, the quantum yield depends on the probability of non-radiative transitions, and, therefore, on the overall mobility of the fluorogen inside the FAP cavity. It is quite difficult to analyze the mobility of a fluorogen, however, one can easily measure the

mobility of the overall complex, and try to minimize it by fixing the fluorogen. Here, we measured the NMR relaxation parameters of [15]N nuclei, to assess the mobility of the FAST backbone in a complex with N871b. In addition, several molecular dynamics (MD) simulations were carried out to investigate the flexibility of protein side chains (Supplementary Fig. 1). The structure of the complex was obtained previously by NMR[16], N871b was selected as a parent fluorogen for the family of compounds, designed for the multicolor labeling[9]. Analysis of the relaxation parameters with a model-free approach reveals three major mobile regions in close proximity to the ligand - loops 98-102, 70-73, and 52-53 (Fig. 1a). There are several other high-mobility sites in the protein, including the N- and C-termini, helices H1, H2, and H5, loops 86-88, 112-116, and a row of residues in the β-strand (Q41, I31, W119, M109). Results of MD simulations are in good agreement with NMR data and reveal the fast sidechain motions in the regions 50-54, 68-74, and 97-102 (Fig. 1b). In close proximity to the ligand, we observed the motions of R52, P68, I96, T98, and S99 sidechains. Thus, we searched for possible mutations in these three regions that could stabilize the complex.

### Design of the enhanced mutants.
It is noteworthy that N871b is substantially different from the initially proposed FAST ligands. The latter are based on 4-hydroxy-benzylidene-rhodanine[1], while N871b contains an additional double bond and a pyridine moiety (Supplementary Fig. 2). This pyridine ring exits the cavity of FAST and is not in tight contact with the protein. Moreover, the pyridine group demonstrates the elevated root-mean-square fluctuations of the atomic coordinates during MD simulations, which may be ascribed to the high propensity of its random flips. On the other hand, due to the peculiar charge distribution (Fig. 2c), this group is capable of various stabilizing interactions: stacking, π-cation, π-anion and hydrophobic. Therefore, we decided to take advantage of these facts and to search for the FAST residues both in the abovementioned dynamic "hot spots" of the protein and in close proximity to the N871b pyridine group. The first and obvious candidate is R52. The residue is in direct contact with pyridine of N871b, but the interaction could be unfavorable. We mutated it to the aromatic amino acids (R52Y and R52F), hydrophobic (R52L), negatively charged (R52E and R52D), positively charged (R52K) and neutral residue (R52A). Following similar principles and taking into account the charge distribution over N871b, we mutated S99 (S99K, S99R, and S99E), P68 (P68K, P68R). We assumed that D65 could stabilize the R52 sidechain via a salt bridge and mutated this residue to either destroy or, conversely, form a salt bridge, when R52 is mutated into a negatively charged residue (D65R and D65K, D65R/R52E). Next, we took several mutations, which proved favorable for other fluorogens[12,14], namely, F62L, P68T, V107I, P73S. Finally, we considered a possibility to change the structure of the loop 98-102 and close the binding pocket by mutating P97 (P97T, P97T/T98G). This led us to 21 mutants, summarized in Supplementary Table 1 and Fig. 2.

### Fluorogenic properties of the FAST mutants.
As a next step, we synthesized in E. coli and purified all the mutant variants, and studied their properties in complex with N871b fluorogen (Fig. 2d, Supplementary Figs. 3–8, Table 1, Supplementary Table 2). Out of 21 tested proteins, only two performed better than the wild-type—R52Y and R52L. Mutations increased both the extinction and the quantum yield and provided ~20–25% higher overall brightness. These two mutants were brighter than the FAST both in the PBS buffer, which was taken to mimic the physiological conditions and in the buffer used for NMR measurements (Supplementary Table 3, Supplementary Fig. 6). Several mutants appeared stabilizing and decreased the dissociation constant but did not improve the

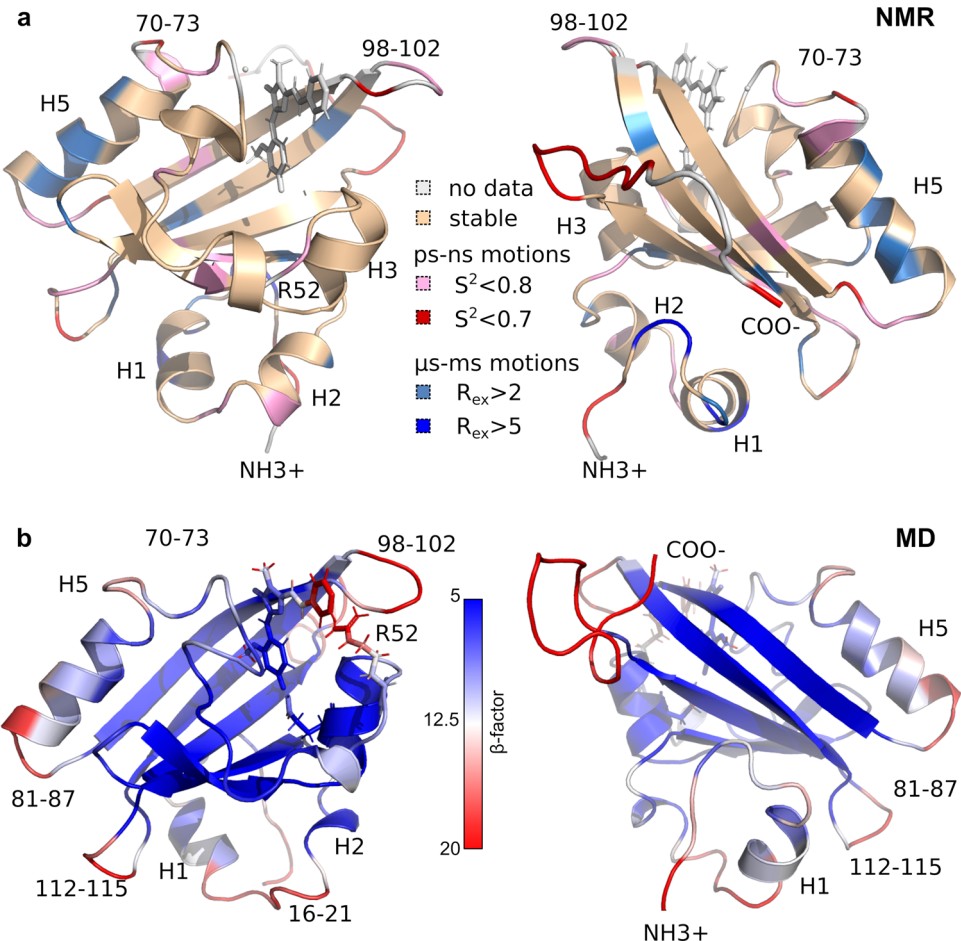

**Fig. 1 Dynamics of FAST/N871b complex. a** The NMR-derived spatial structure of FAST/**N871b** is colored according to the generalized order parameters ($S^2$) of NH bonds (pink and red, ps-ns motions) and exchange contributions to the transverse relaxation ($R_{ex}$, blue, µs-ms motions). **b** The resulting FAST/**N871b** structure after 100 ns MD simulations is colored according to the magnitudes of atom B-factors. Blue and red correspond to relatively stable and mobile regions, respectively.

brightness of the complex (all the S99 mutants, P73S, P68K, D65K). Finally, several suggested mutations completely destroyed the fluorogen-activating ability of FAST with respect to **N871b** (P97T and R52D). Therefore, we can conclude that the initial suggestion of the prospective mutation sites was correct, R52, D65, P68, and S99 mutations either improved the brightness or improved the binding affinity of FAST. However, it is not sufficient to find a mutation that would decrease the free energy of the FAST/**N871b** complex. Apparently, it is also necessary to reduce the conformational lability of the chromophore by imposing restrictions on its particular degrees of freedom that determine the spectral properties. These restrictions are quite difficult to predict, which explains that only one of selected mutagenesis sites provided a reasonable improvement in brightness.

To find out whether R52Y and R52L mutations can improve the performance of FAST in complex with other fluorogens, we tested three structural analogs of **N871b**, previously proposed for multicolor labeling: **MCL.1b** (emission at 579 nm), **MCL.1f** (593 nm) and **MCL.3a** (639 nm)[9] (Table 1, Fig. 2d, Supplementary Figs. 9–16). The classical FAST ligand **HMBR** was also added to this set as a sample compound with drastically different structure. Brightness of all three **N871b** analogs was increased in complexes with both mutants in comparison to the parent protein. The greatest enhancement was observed for **MCL.1b** and **MCL.1f** in complexes with FAST-R52Y (25 and 27%, respectively) and for **MCL.3a** in a complex with FAST-R52L (34%). In

contrast, **HMBR** was much less bright in complexes with the mutant proteins, the brightness decreased by 36 and 47% for R52L and R52Y, and the stability of the complex decreased by one order of magnitude. Thus, the effects of R52Y and R52L mutations in FAST are ligand-specific and provide the substantial enhancement for several arylidene-imidazolones based fluorogens, applicable for the multicolor fluorescent labeling. With this regard we suggest naming the optimized variants of FAST as (multicolor) mcFAST-Y and mcFAST-L.

Next, we compared the optimized FAST variants and original FAST with various fluorogens in live-cell experiments. We used a TagBFP2 fluorescent protein as an internal reference, which was included in the H2B-TagBFP2-FAST construct, that allowed us to estimate the relative brightness (Supplementary Figs. 17, 18). As clearly seen in Fig. 3, the values of relative brightness of all optimized pairs were in good agreement with in vitro data. Besides, we measured the photostability of FAST variants with the chromophore **N871b**. The resulting curves indicate that there is no difference in photostability between them (Supplementary Fig. 19). Some discrepancies probably arose because the wavelength of the exciting light in microscopy did not always correspond to the maximum absorption for which the EC was measured and the relative brightness was calculated. However, mcFAST-Y turned out to be significantly brighter in pair with fluorogens **N871b**, and **MCL.1b**, while mcFAST-L is brighter with fluorogen **MCL.3a**.

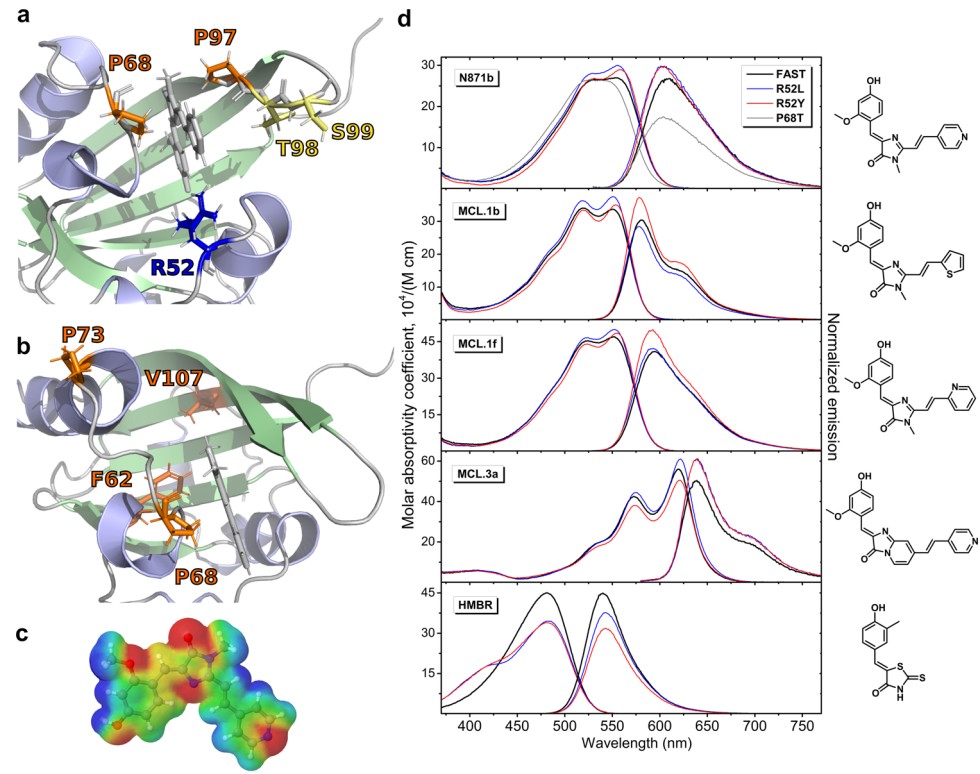

**Fig. 2 Mutagenesis of FAST and absorption/emission spectra of brightest mutants. a** Residues, rationally selected for mutagenesis in the current study as the closest to the pyridine moiety of N871b (shown with letters and numbers). **b** Residues that were mutated, because their substitutions proved favorable for other fluorogens in the literature. **c** Electrostatic potential distribution over the contact surface of **N871b** according to DFT calculations. Red zones correspond to the negative potential, blue - to the positive one. **d** Absorption and emission spectra of various fluorogens in complexes with FAST protein and its brightest mutants. Absorption spectra are normalized to the protein/ligand complex concentration and represented in the molar absorptivity coefficient scale. Emission spectra are normalized to the FQY.

**Table 1 Optical properties of HMBR, N871b, MCL.1b, MCL.1f and MCL.3a in complexes with FAST mutants.**

| Chr. | Mutant | $K_D$, μM [a] | ε, $M^{-1} \cdot cm^{-1}$ [b] | FQY, % [c] | Brightness |
|---|---|---|---|---|---|
| HMBR | FAST | 0.13 ± 0.02 | 45,000 ± 680 | 31 ± 0.6 | 13,900 ± 480 |
| | mcFAST-L | 0.48 ± 0.11 | 34,500 ± 520 | 26 ± 0.6 | 8900 ± 340 |
| | mcFAST-Y | 1.34 ± 0.11 | 34,000 ± 510 | 22 ± 0.1 | 7350 ± 140 |
| N871b 609 nm | FAST | 0.33 ± 0.01 | 27,000 ± 410 | 26 ± 1.4 | 7000 ± 480 |
| | mcFAST-L | 0.27 ± 0.01 | 30,000 ± 450 | 29 ± 1.7 | 8750 ± 640 |
| | mcFAST-Y | 0.24 ± 0.02 | 29,000 ± 440 | 29 ± 1.5 | 8350 ± 560 |
| | pFAST | 0.040 ± 0.002 | 28,500 ± 430 | 31 ± 1.0 | 8850 ± 420 |
| MCL.1b 582 nm | FAST | 0.20 ± 0.04 | 34,000 ± 510 | 31 ± 2.2 | 10,650 ± 920 |
| | mcFAST-L | 0.44 ± 0.20 | 37,500 ± 560 | 29 ± 0.6 | 11,000 ± 390 |
| | mcFAST-Y | 0.27 ± 0.06 | 35,000 ± 530 | 38 ± 1.4 | 13,350 ± 690 |
| | pFAST | 0.023 ± 0.003 | 37,000 ± 560 | 34 ± 1.8 | 12,500 ± 850 |
| MCL.1f 595 nm | FAST | 0.27 ± 0.04 | 47,000 ± 710 | 32 ± 0.6 | 14,800 ± 500 |
| | mcFAST-L | 0.39 ± 0.10 | 50,000 ± 750 | 33 ± 1.7 | 16,500 ± 1100 |
| | mcFAST-Y | 0.31 ± 0.05 | 48,500 ± 730 | 39 ± 1.3 | 18,850 ± 910 |
| | pFAST | 0.029 ± 0.005 | 50,500 ± 760 | 37 ± 1.2 | 18,850 ± 890 |
| MCL.3a 639 nm | FAST | 0.08 ± 0.01 | 56,000 ± 840 | 14 ± 0.7 | 7900 ± 510 |
| | mcFAST-L | 0.06 ± 0.01 | 61,000 ± 920 | 17 ± 0.6 | 10,550 ± 530 |
| | mcFAST-Y | 0.11 ± 0.02 | 50,500 ± 760 | 17 ± 1.4 | 8300 ± 810 |
| | pFAST | 0.025 ± 0.002 | 58,500 ± 880 | 14 ± 1.8 | 8200 ± 1180 |

a – represented as mean ± SD ($n = 3$);
b – represented as result of single measurement ± the precision of the measuring instruments (weighing and pipetting errors);
c – fluorescence quantum yield, represented as mean ± SD ($n = 9$, except **HMBR**-mcFAST-Y, where $n = 3$).

As we have mentioned in the Introduction, there are multiple enhanced FAST variants, obtained by the directed evolution. One of the most prospective is pFAST[10]—the promiscuous variant, capable of enhancing the fluorescence of various fluorogens. To test, how our rationally designed single-point mutants behave in comparison to pFAST, we synthesized this protein and measured its brightness in complexes with all four ligands, investigated in the present work (Table1, Supplementary Figs. 9–16). In all the

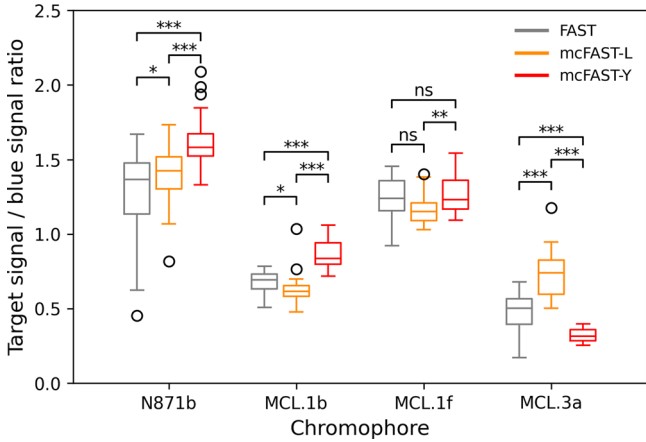

**Fig. 3 In-cell brightness of FAST and its mutants.** Comparison of relative brightness of FAST mutants with different chromophores in live HeLa Kyoto cells transiently transfected with H2B-TagBFP-FAST(or its mutants) constructs. Whiskers show standard deviation, horizontal lines indicate median values, and circles indicate outliers. $n = 29$ for each box. Stars indicate the $p$-values: * $p < 0.05$, ** $p < 0.01$, *** $p < 0.001$, ns $p > 0.05$.

cases, pFAST demonstrated a significantly higher binding affinity than FAST, mcFAST-L and mcFAST-Y. However, the brightness of pFAST was comparable but not superior to the proteins, proposed in the present work. In the case of **MCL.3a**, pFAST did not provide a substantial brightness enhancement, while mcFAST-L did. Thus, the rational approach allowed constructing a FAST mutant with the brightness, comparable or superior to pFAST, found by the directed evolution as a protein, optimized for a wide variety of ligands.

**Structural determinants of fluorogen-activating ability of FAST mutants.** The obtained set of mutants with varied brightness could be used to find out the structural and dynamic features of the complex that determine the performance of FAST protein. Solving the spatial structure and measuring the backbone mobility for all 21 FAST variants is too expensive and time-consuming, however, some important features could be found even in 1D NMR spectra. Namely, the downfield region of $^1$H spectrum of the FAST/**N871b** complex contains three protons that take part in intermolecular hydrogen bonding - Y42 Hη, E46 Hε2 (with the oxygen of phenolic ring), and Hε1 W94 (with the oxygen of imidazolone ring) (Fig. 4a–f, Supplementary Fig. 20, Supplementary Table 4). Thus, we characterized the hydrogen bonding in the studied FAST/**N871b** complexes, using the proton chemical shift as an indicator of the H-bond length[17], and normalized intensity of the signal as an indicator of the H-bond stability. The correlation analysis reveals that the overall brightness of the complex demonstrates a correlation with the chemical shift of W94 ($r = 0.57$, $p < 0.01$) and a weak correlation with the chemical shift of E46 ($r = 0.40$, $p < 0.1$). The greater the chemical shift, the shorter the corresponding H-bond, the brighter the complex. Correlation between the "stability" of three hydrogen bonds (sum of the normalized intensities of three observed signals) and brightness is also weak: $r = 0.36$, $p < 0.2$. However, we observe three highly stable mutants: mcFAST-L, mcFAST-Y, and R52F, two of which provide substantial improvement. Therefore, high H-bond stability is a good indicator of the potentially high brightness of the FAST/**N871b** complex. A more detailed analysis with separate treatment of various contributions reveals the reliable correlations between the W94 chemical shift and the quantum yield, as well as between the stability of E46 and Y42 H-bonds and the extinction coefficient

(Fig. 4e). To summarize, the brightest mutants correspond to the complexes with the shortest intermolecular H-bonds formed by W94 side chains and the highest stability of intermolecular H-bonds formed by E46 and Y42. However, there are outlying mutants - R52A, which is brighter than FAST, but does not reveal any NMR properties of brightest mutants, and R52F, which demonstrates the characteristics of the bright mutant, but is dimmer than the original FAST tag. This, together with the rather weak character of correlations, suggest that there are other factors that affect the brightness of FAST/**N871b** complex that we still do not understand.

The structural basis of the enhancement could be analyzed with MD simulations as well. For this purpose, we ran MD simulations with the parent and several mutant FAST proteins, starting from the NMR structure (Fig. 5a). In the parent protein, the sidechain of R52 tends to form a π-cation interaction with the pyridine moiety of **N871b**. The same behavior is observed in the case of mcFAST-Y: the tyrosine sidechain forms stacking interaction with the pyridine ring. However, the R52 sidechain has many degrees of freedom, which makes it mobile and allows the conformational diversity (Fig. 5a, Supplementary Fig. 20). In contrast, tyrosine seems to have only two possible modes of stacking. This restricts the motions of Y52 in mcFAST-Y and may be considered as a source of **N871b** stabilization and fluorescence enhancement. The behavior of leucine at position 52 is radically different. This sidechain does not interact with the pyridine group of the ligand - it is tightly packed together with the aliphatic chains of V57 and V66. These hydrophobic interactions restrict the mobility of L52 - it is locked mostly in a single $\chi_1/\chi_2$ conformation (Supplementary Fig. 21). Thus, the source of the enhancement in the case of mcFAST-L is the overall stabilization of the protein core via the favorable residue packing.

To investigate the structural properties of the brightest mutants in more detail, we synthesized the $^{15}$N-labeled mcFAST-Y, assigned its backbone chemical shifts and investigated the intermolecular mobility of the mcFAST-Y/**N871b** complex. In agreement with MD simulations, the structure of mcFAST-Y did not change substantially, the drastic chemical shift perturbations were observed only in the immediate vicinity of the Y52 residue that was mutated (Supplementary Fig. 22). The overall pattern of internal backbone mobility remained similar to FAST as well (Supplementary Fig. 23), while the backbone order parameters of residues 52 and 53 increased (Fig. 5b), thus supporting the stabilization of this "hot spot" by the introduced mutation. Besides, we observed the decreased contributions of slow μs-ms motions to the transverse relaxation of mcFAST-Y, both at the N-terminus and in the H5 helix (Fig. 5c).

To have a complementary data, we additionally measured the thermostability of FAST, mcFAST-Y and mcFAST-L using the nanoDSF approach. As we found out, the melting of proteins in the presence and in the absence of **N871b** occurs at exactly the same temperature, suggesting that complexes dissociate at lower temperatures, probably due to the temperature dependence of the dissociation constant and limited water solubility of the fluorogen (Supplementary Fig. 24). In other words, we are able to detect only the melting of the protein apo state, and the obtained data reveal that this ligand-free form is stabilized by both the R52Y and R52L mutations. While R52Y increases the melting temperature by $1.4 \pm 0.2°$, R52L provides the stabilization by $3.5 \pm 0.2°$ (Supplementary Fig. 24).

To summarize this part, we observe that the brightest FAST mutants are characterized by improved stability, which is manifested in multiple aspects of their behavior, from the length of intermolecular H-bonds to the local decrease of specific motions and increase of the melting temperature of the protein apo state.

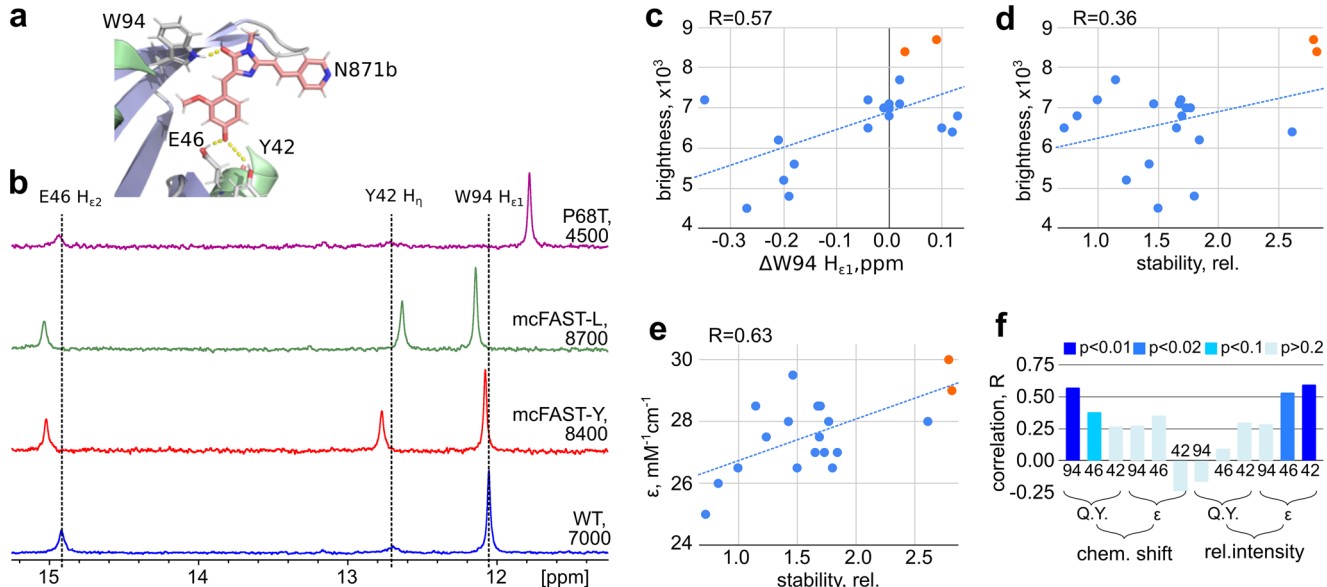

**Fig. 4 NMR analysis of FAST/N871b complexes. a** Ligand/protein intermolecular hydrogen bonds are shown on the structure of FAST/N871b complex. **b** Overlay of $^1$H NMR spectra of FAST/**N871b** complexes, recorded for the wild-type protein and its three mutants: mcFAST-Y, mcFAST-L (the best mutants) and P68T (the worst mutant). Spectra were obtained at 25 °C, molecular brightness of the complex is indicated. **c** Correlation between the chemical shift of Hε1 W94 and brightness of FAST/**N871b** complex. **d, e** correlations between the FAST/**N871b** brightness and NMR-derived stability of three intermolecular H-bonds (**d**) or molar extinction coefficient (ε, **e**). mcFAST-Y and mcFAST-L mutants are highlighted in orange. **f** Pearson correlation coefficients, obtained for the chemical shifts and relative intensities of signals, corresponding to intermolecular H-bond protons of FAST/**N871b** complexes, and either quantum yield (Q.Y.) or molar extinction coefficient. Bars are colored according to the statistical significance of the correlation.

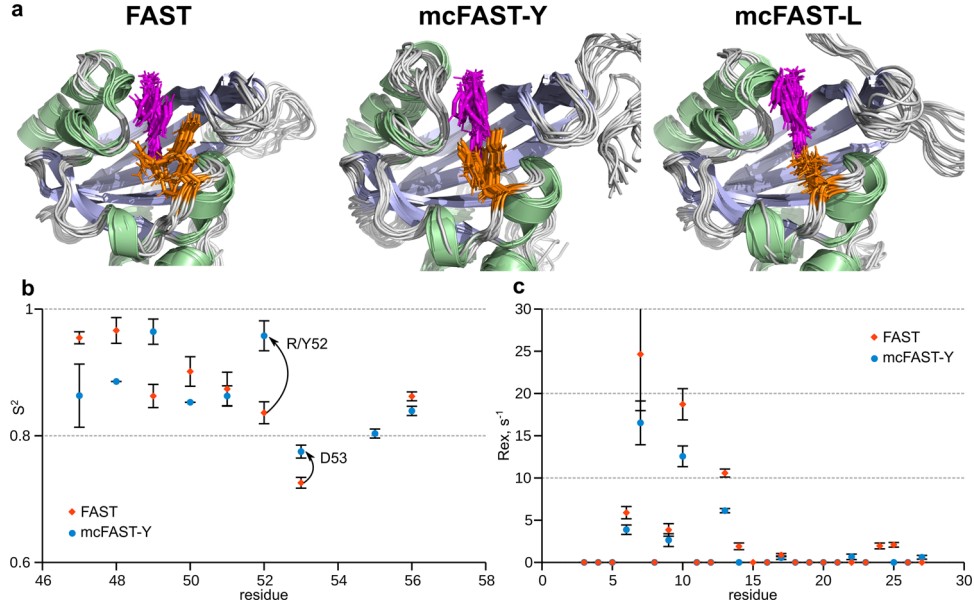

**Fig. 5 Dynamics of FAST brightest mutants. a** 11 structures of FAST/**N871b** complex (FAST, mcFAST-Y and mcFAST-L mutants), selected from the last 10 ns of the MD simulation in 1 ns interval. **N871b** is shown in magenta and the sidechain of residue 52 is shown by orange sticks. Structures are superimposed over the backbone atoms of the β-sheet residues. **b** Generalized order parameters S$^2$ are shown for residues 47-56 of FAST (blue circles) and mcFAST-Y (orange rhombuses), according to the model-free analysis of $^{15}$N relaxation parameters. Changes, observed for residues 52 and 53 are shown by arrows. **c** Contributions of slow µs-ms motions to the transverse relaxation of the N-terminal residues (3-27) of FAST (blue circles) and mcFAST-Y (orange rhombuses). Error bars in panels **b** and **c** correspond to the errors of parameters of approxiamtion, determined by the Monte-Carlo analysis.

## Discussion

In the present work, we suggest a rational approach to the modification of FAST protein for the particular fluorogen **N871b**. The approach relies on the analysis of protein dynamics with NMR and computer simulations. Such an analysis provides the "hot spots" that may be improved, aiming to freeze the motions of both protein and fluorogen. According to our data, mobile regions include the N-terminal two-helical domain (1-26), which becomes unstructured in the apostate of FAST[16], five loops: 50-54, 70-74, 85-89, 97-103, and 111-117. Slow µs-ms motions are also observed in the helix H5 (Fig. 1). In case these "hot spots" are common for various ligands, the FAST optimization strategy

should include the mutations that decrease mobility in the specified regions. To test whether the found regions represent the common mutagenesis sites for different ligands, we analyzed the known mutations in FAST that were previously reported to enhance the brightness or ligand-binding efficacy of the protein. To date, 58 FAST mutants were published as the final or intermediate result of optimization, including the iFAST, rFAST, gFAST, frFAST, oFAST, tFAST, and pFAST[10–13]. These 58 proteins contained in total 253 mutations, 97 substitutions were unique and were located at 54 different sites (Table S5). The largest fraction of this set includes 26 mutations that affect the N-terminal domain or its interactions with the FAST core, 14 mutations affect the residues that are in direct contact with the ligand molecule and 15 mutations are located in helix H5. Finally, 23 substitutions are located in the loops that were found mobile in the current study. Therefore, the vast majority of known mutations (64 of 97) are located in the "hot spot" regions, 54 out of 58 mutant proteins contained the mutations in the "hot spots", which supports the general character of our findings (Fig. 6).

Using the proposed approach, we suggested 17 mutants, 2 of which (mcFAST-Y and mcFAST-L) revealed the improved brightness, corresponding to the success rate of ~12%. These mutations appeared beneficial not only for N871b, but also for several similar compounds with various emission maxima. The enhancement (20-34%) is comparable to the effect observed in the case of iFAST$_{V107I}$ (10-30% for various fluorogens), which was also found by rational design based on the structure of PYP protein, parent to FAST[12]. Compared to directed evolution, rational design has its advantages and drawbacks. Fast screening approaches, used for selection of clones, are not very precise. Therefore, if the gain of function is not substantial, it may be left unnoticed and the enhanced variant would be dismissed. Besides, the brightness in the living cells may depend on the fluorogen binding constant and the expression level of the protein, which may result in the selection of variants that are not brighter, when the protein is purified and analyzed in vitro. This was clearly demonstrated in the case of three FAST mutants that were selected in massive screening as the brightest[14], but were not superior to FAST, when isolated[12]. On the other hand, using the rational approach it is hard to predict and construct the

multipoint mutants, such as pFAST[10], which provided the 2-3 fold enhancement relative to FAST for some fluorogens. Nevertheless, the direct comparison of mcFAST-Y and mcFAST-L with pFAST revealed that the latter is not superior to the rationally designed single-point mutants, when the brightness with the fluorogens based on N871b is analyzed. Thus, today, rational design may be considered as a feasible alternative to the directed evolution, when the FAP is well adapted to the fluorogen and only a fine-tuning is required.

To conclude, here we report a successful attempt to rationally design an enhanced fluorogen-activating protein mutant for a particular fluorogen. Using the spatial structure of the FAST/N871b complex, NMR relaxation parameters, and MD-simulations, we identified the mobile regions in the complex and rationally suggested 17 mutations that could stabilize both the protein and the pyridine group of N871b. Two of our mutants appeared superior to the wild-type FAST, while none of the tested mutations, that were shown favorable for FAST in complex with other fluorogens, provided any improvement for N871b. The mutants selected for N871b performed well for several other fluorogens of similar structure and did not work with conventional FAST fluorogen HMBR. Analysis of the set of mutants by NMR reveals that brighter mutants demonstrate the highest stability and the shortest length of intermolecular H-bonds and MD simulations provide a structural basis for such a stabilization. According to the literature, most of the known FAST mutations that were found in numerous directed evolution studies are located in the mobile regions reported here. This suggests the general character of our findings and provides the basis for rational combination of mutations that stabilize the protein and mutations that affect ligand binding.

## Methods

**General**. All solid chromophores were dissolved in DMSO (Sigma Aldrich, "for molecular biology" grade, #cat D8418) in 5 mM concentration, and stored in a dark place at −20 °C for no more than 3 months.

UV-VIS spectra were recorded on a Varian Cary 100 spectrophotometer. Fluorescence excitation and emission spectra were recorded on Agilent Cary Eclipse fluorescence spectrophotometer.

Absorption and emission maxima are determined as is from raw data, without using any approximation procedure.

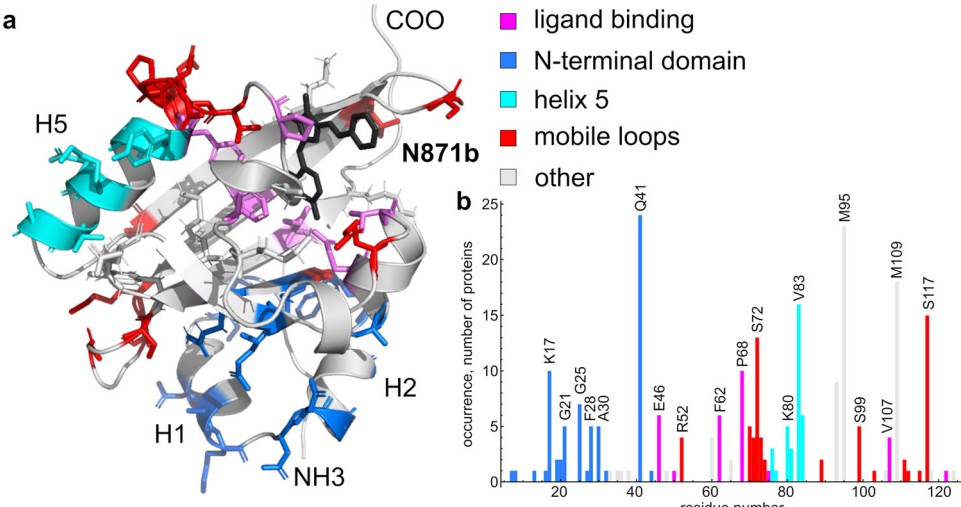

**Fig. 6 Analysis of previously published FAST mutations. a** Structure of the FAST/N871b complex. Sidechains of residues that were mutated in any of the enhanced FAST variants are shown by sticks and colored with respect to the presumed role of the mutation. Ligand is shown in black. **b** Occurrences of mutations at certain positions in 58 reported enhanced FAST variants. In both panels, magenta corresponds to the residues, directly involved in the ligand binding; blue - to the residues that support the packing of the N-terminal domain; cyan - to the residues in the helix H5, red - to the residues in mobile loops. Gray denotes the mutations that do not fall in any of the listed groups. Figure is based on the data provided in the Supplementary Table 5.

**Protein production**. All genetic constructs based on pET24b(+) and pEV expression vectors were produced by Cloning Facility (Russia) and their identity was confirmed by DNA sequencing. The sequence encoding His6-tag was added at the 3' end of target protein genes via several nucleotides depending on the vector and cloning procedure used. The resulting sequences were the following: KLAAALEHHHHHH in pET24b(+) for the FAST WT used for NMR spectroscopy and GGGHHHHHH in pEV for pFAST, FAST WT and mutants used for all the experiments. The behavior of both variants of FAST WT were identical.

The proteins were produced and purified similar to the protocol described earlier for the FAST protein[16]. In brief, the BL21(DE3) E.coli strain and M9 minimal salts medium were used for protein production. In order to obtain the $^{15}$N-labeled protein samples, $^{15}$NH$_4$Cl was implemented. Cells were grown at 37 °C, 250 rpm (300 mL of cell culture per 2 l Ehrlenmeyer flask) in a shaker incubator (New Brunswick Innova 44 R). Protein expression was induced at OD$_{600}$ ~ 0.6 using 0.25 mM of isopropyl β-d-1-thiogalactopyranoside (IPTG). After 4 hours of cultivation cells were harvested and stored at −20 °C. Cells were lysed by sonication (Bandelin Sonoplus, GM 2200, titanium tapered tip KE76) on ice followed by target protein purification using Ni$^{2+}$ Chelating Sepharose High Performance (GE Healthcare) and overnight dialysis (12 kDa MWCO) against the 1xPBS buffer at room temperature (Supplementary Fig. 25).

**NMR spectroscopy**. NMR spectra for the relaxation analysis were recorded in NMR buffer (pH 7.0, 20 mM NaPi, 20 mM NaCl). The concentration of protein was 0.85 mM(FAST) or 0.90 mM (mcFAST-Y), the concentration of **N871b** was equal to 0.89 mM/1.0 mM, respectively. The experiments were run at 25 °C using the Bruker Avance 700 MHz spectrometer. $^{15}$N longitudinal (T1), transverse (T2) relaxation rates were measured using the pseudo-3D HSQC-based experiments with varied relaxation delay[18]. Heteronuclear equilibrium $^1$H,$^{15}$N-NOE magnitudes were obtained using the $^1$H presaturation for 3 s during the recycling delay. Reference and NOE spectra were recorded in the interleaved mode. Relaxation parameters were analyzed using the model-free approach as implemented in the TENSOR2 software, assuming the isotropic rotation[19]. Assignment of mcFAST-Y/ **N871b** backbone was performed based on the FAST assignment that was obtained previously, using the 3D $^{15}$N-NOESY-HSQC and $^{15}$N-TOCSY-HSQC spectra.

To measure the $^1$H spectra, FAST or its mutants in PBS buffer pH 7.4 were mixed with **N871b** at 10% molar excess of fluorogen. The resulting concentrations of protein were 50-300 μM. NMR spectra were recorded at 25 °C using the Bruker Avance III 800 MHz NMR spectrometer, equipped with a triple-resonance cryoprobe. $^1$H chemical shifts were referenced with respect to the water signal, chemical shift of water protons was taken equal to 4.80 ppm at 25 °C. Relative intensities of $^1$H signals were calculated as 1/3 of a ratio between the integral of the signal of interest and of an alone-standing methyl group at −0.66 ppm. The "stability" of an intermolecular H-bond is calculated as a ratio between the relative intensity of the signal of interest and the maximal relative intensity, obtained for this signal in the whole set of tested mutants. The summary stability of three intermolecular H-bonds is calculated as a sum of three individual stabilities and lies in the range {0;3}.

**Computer simulations**. Five starting structures of the complex (models 1, 5, 9, 13, 17) were chosen from pdb entry 7AVA[16]. Protein mutations were made using the mutagenesis wizard of the PyMOL Molecular Graphics System (Schrödinger LLC). MD simulations were carried out in Gromacs version 2020.4 software[20]. All-atom Amber99sb-ildn[21] forcefield was used for protein, tip3p model for water and parameters for **N871b** were developed in this work. Equilibrium geometry (minimum energy conformation) of **N871b** and energy profiles for five dihedral angles ($\chi_1$ - $\chi_5$, Supplementary Fig. 26) were calculated using DFT (B3LYP functional, 6-31 G(d,p) basis set) in GAMESS software[22,23]. Molecular electrostatic potential for **N871b** was visualized from these calculations using JMol program.

Atom types, bond and angles parameters were assigned in Ambertools 20[24]. Partial atomic charges were calculated by averaging DFT charges of chemically equivalent atoms. Since energy profiles have clear minima in planar conformations (Supplementary Fig. 27) we had to exclude some nonbonded interactions that shift the minimum position. Excluded nonbonded atom pairs are shown in Supplementary Fig. 25. Parameters for dihedral angles $\chi_2$, $\chi_3$, and $\chi_5$ were calculated by fitting an energy barrier between two minima in GAMESS and Gromacs energies with proper periodic dihedral angle terms. Dihedral angle energy barriers for angles $\chi_1$, and $\chi_4$ are extremely large and prevent flips between two minima. To correctly describe these dihedral angle parameters around their minima we used Ryckaert-Bellemans dihedral potential. Ryckaert-Bellemans parameters were calculated by fitting GAMESS and Gromacs energies (Supplementary Fig. 27).

Starting structures were placed in the center of the dodecahedron box at least 1.2 nm from the box edge, solvated with water. Na$^+$ Ions were added to neutralize the system charge. All simulations were performed with periodic boundary conditions, Verlet cutoff scheme, plain cutoff of 1.4 nm for van-der-Waals and particle-mesh Ewald method for Coulomb interactions. Simulation included 4 stages: energy minimization in double precision, heating from 5 to 300 K during 1 ns with 1 fs timestep using V-rescale thermostat and fixed protein, equilibration during 50 ns with 2 fs timestep using V-rescale thermostat and Parrinello-Rahman barostat, production run 200 ns with the same parameters. Lincs algorithm was employed to constrain the bonds with H-atoms in MD simulations except for the

heating phase. RMSD from starting structures (Supplementary Fig. 28) showed that 50 ns is obviously enough for system equilibration.

**Measurement of dissociation constants**. The experiments were performed on the Tecan Infinite 200 Pro M Nano dual mode plate reader at 25 °C. The affinity constants were determined by spectrofluorometric titration of protein in single concentration (0.10 μM) using various chromophore concentrations. Fitting was performed using Origin 8.6 software. Titration curves are presented in Supporting Information (Supplementary Figs. 3–6 and 9–12).

**Measurement of extinction coefficients**. The chromophores solutions were mixed with a protein solution in PBS buffer (pH 7.4, #cat E404-200TABS, Amresco) or NMR buffer. The final concentration of chromophores (together in free form and in complex) for all experiments was 5 μM. Proteins were added in such amounts that led to almost complete transfer of the chromophore into the complex ($\alpha \geq 97\%$). The final protein concentration (together in free form and in complex) for each complex was calculated using the equation:

$$[Pr] = \frac{K_D \times (\alpha \times [Chr])}{[Chr] - (\alpha \times [Chr])} + (\alpha \times [Chr]) \tag{1}$$

where $K_D$ – dissociation constant, [Chr] – final chromophore concentration.

The molar extinction coefficient was calculated by the formula:

$$\varepsilon = \frac{A}{cl} \tag{2}$$

where $A$ is the absorbance intensity at maxima, $c$ is the molar concentration of complexes, $l$ is the pathlength.

The data were obtained from a single measurement. The errors were determined from the precision of the measuring instruments (weighing and pipetting errors).

**Measurement of quantum yields**. Fluorescence quantum yields were calculated according to the procedure described in the literature[25] with the use of Rhodamine 101 (for **N871b**, **MCL.1b** and **MCL.1 f**), Fluorescein (for **HMBR**) and Oxazine 1 (for **MCL.3a**) as standards. The final concentration of chromophores (together in free form and in complex) was 5 μM for absorption spectra registration and 1.67, 0.5, and 0.167 μM for emission spectra registration, (except R52E-**N871b**, R52E/ D65R-**N871b** and R52Y-**HMBR** pairs, where only 0.5 μM chromophore concentration was used for emission spectra registration). Proteins were added in such amounts that in all cases lead $\alpha \geq 97\%$ (see above).

The quantum yield was calculated by the formula:

$$\Phi_x = \Phi_{st} \times \frac{F_x}{F_{st}} \times \frac{f_{st}}{f_x} \times \frac{n_x^2}{n_{st}^2} \tag{3}$$

where F is the area under the emission peak, f is the absorption factor (see below), n is the refractive index of the solvent, Φ is the quantum yield, the subscript x corresponds to the complexes, the subscript st – for standards.

$$f = 1 - 10^{-A} \tag{4}$$

where A is absorbance at the excitation wavelength.

**Molecular cloning**. Plasmids for mammalian expression of H2B-TagBFP fusions with FAST and FAST mutants (mcFAST-L, mcFAST-Y) were assembled using Golden Gate cloning following MoClo standard[26]. Each transcriptional unit for mammalian expression consisted of the CMV promoter, coding sequence for the fusion protein, and the SV40 terminator. All Golden Gate cloning reactions were performed in the T4 ligase buffer (ThermoFisher, USA) supplied with 10 U of T4 ligase, 20 U of either BsaI or BpiI (ThermoFisher, USA), and 100 ng of DNA of each DNA fragment. Golden Gate reactions were performed with the following cycling conditions: 30 cycles between 37 °C and 16 °C (90 sec at 37 °C, 180 sec at 16 °C).

**Cell culture and transient transfection**. HeLa cells were grown in Dulbecco's modification of Eagle's medium (DMEM) (PanEco) supplied with 50 U/ml penicillin and 50 μg/ml streptomycin (PanEco), 2 mM L-glutamine (PanEco), and 10% fetal bovine serum (HyClone, Thermo Scientific) at 37 °C and 5% CO2. Cells were plated and grown in 35-mm glass-bottom dishes (SPL). For transient transfection, FuGENE HD reagent (Promega) was used. Imaging was performed after 24-26 h after transient transfection. Immediately before imaging DMEM was replaced with Hanks Buffer (PanEco) supplemented with 20 mM HEPES (Sigma).

**Fluorescent microscopy**. Widefield fluorescence microscopy of live HeLa cells was performed with Leica DMI6000b inverted microscope equipped with HCX PL FLUOTAR L 20×0.40 CORR lens, CoolLED pE-300 light source, Andor Zyla 5.5 sCMOS camera (Andor), and using Blue, mCherry, and CY5 filter cubes. Illumination powers in blue, red and far-red channels were 0.47 W/cm$^2$, 0.8 W/ cm$^2$, and 0.2 W/cm$^2$ respectively. Cells were imaged before and after chromophore addition to the final concentration of 20 μM for each chromophore.

Photobleaching was carried out using Nanoimager S (Oxford Nanoimaging). As a light source 561 nm laser was used. Illumination power for photobleaching was 0.7 kW/cm$^2$ of 561 nm laser light.

**Relative brightness calculation**. For evaluation of relative brightness of FAST mutants with chromophores, we selected regions of interests (ROI) in cell nuclei. Then the target signal in red of the far-red channel in each ROI was divided by the signal in the channel of the TagBFP fluorescent protein. Obtained values of the relative brightness were used to draw boxplots. To obtain p-values a two-tailed t-test was used.

**NanoDSF measurements**. The thermal stability of FAST, mcFAST-Y and mcFAST-L, both in the apo state and in the complex with **N871b** was measured by nanoDSF using Prometheus Panta in nanoDSF grade standard capillaries (Nano-Temper Technologies GmbH, München, Germany). The thermograms were measured from 25 to 95 °C at the ramping rate of 2 °C/min with the protein concentration of 0.11 mg/mL in the PBS buffer containing 2% v/v DMSO, **N871b** was added in 3x molar excess. Fluorescence was excited at 280 nm and the emission was collected at 330 nm and 350 nm with dual-UV detector. Thermal stability parameters were calculated by PR.Panta Analysis v1.2 software. Tm was calculated as an inflection point from the changes in the ratio of aromatic amino acids emission at 330 nm and 350 nm, as a function of temperature.

**Statistics and reproducibility**. Dissociation constants of FAST/fluorogen complexes were measured in three independent experiments. Fluorescence quantum yields were measured in nine experiments, for three samples with various concentrations of the protein and at three different wavelengths. In-cell brightness was measured for 29 different cells for each protein/ligand pair, the statistical significance of the differences was assessed using the two-tailed t-test. NanoDSF measurements were performed for three independent sample preparations for each protein tested.

**Reporting summary**. Further information on research design is available in the Nature Research Reporting Summary linked to this article.

## Data availability

All the data are available on request to the corresponding authors. Unedited SDS-PAGE gels are provided as supplementary figures 29 and 30. Raw data that was used for the charts in MEP_L_fig3Figs. 3 and 4 are provided as supplementary data file 1.

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

## Acknowledgements

The work was supported by the Russian Science Foundation grant # 18-73-10105. MD simulations were supported by the HSE University Basic Research Program and carried out with the use of computational facilities of the Supercomputer Center "Polytechnical" at the St. Petersburg Polytechnic University and IACP FEB RAS Shared Resource Center "Far Eastern Computing Resource" equipment (https://cc.dvo.ru). NanoDSF measurements were supported by the Russian Ministry of Science and Higher Education grant # 075-15-2021-1354.

## Author contributions

K.S.M., M.S.B. and R.G.E. designed the experiments, M.V.G. and S.A.G. synthesized the recombinant proteins, K.S.M. performed the NMR experiments, N.S.B. measured the fluorescence efficiency of the mutant proteins, A.Y.S. did the QM calculations, D.E.N. performed the MD simulations, A.V.M and V.I.B. performed NanoDSF measurements, A.S.G. and A.S.M performed the living cell transfection and fluorescence microscopy, K.S.M., M.S.B., R.G.E. and A.S.A. supervised the project, M.S.B. acquired the funding, K.S.M. and M.S.B. wrote the paper with assistance from all the authors.

## Competing interests

The authors declare no competing interests.
