## [Peer Review File · Communications Biology]

Reviewers' comments:

Reviewer #1 (Remarks to the Author):

Mineev, Baranov and coworkers present the development of variants of the fluorescence-activating and absorption-shifting tag (FAST) displaying enhanced properties with the N871b fluorogen and other fluorogens derived from the GFP chromophore. Using the NMR structure of the FAST:N871b assembly, the authors performed NMR relaxation analysis and molecular dynamic simulation to identify mobile regions in the protein-fluorogen assembly, in particular those in close proximity to the ligand. This analysis allowed them to identify putative hot spots that could be mutated in order to stabilise the assembly. Ultimately they analysed 21 rationally designed mutants and could identify that the mutations R52Y and R52L increased the absorptivity and fluorescence quantum yield of FAST:N871b, leading to an overall increase of brightness of 20-25%. The authors also showed that these mutants displayed improved properties with structural analogs of N871b. The authors demonstrated that these variants displayed increased brightness in live cells as well, in agreement with the in vitro data.

The paper is well written and pleasant to read, and the study is well done. The methodology developed to analyse the interaction of FAST variants with various fluorogen by NMR relaxation analysis and MD simulations opens new prospects for characterising FAST:fluorogen pairs. The discovery of new variants with improved properties for N871b and its analogs is interesting as it enables to expand the FAST toolbox. The paper could be however significantly improved by addressing the following points:

- Although I agree that solving the NMR structure and measuring the backbone mobility of all 21 FAST variants analysed in this study was probably not necessary, it would have been very useful to do this analysis at least for the best variant mcFAST-Y (or mcFAST-L) in order to demonstrate the effect of the mutations and support their rational approach. Addition of such analysis would significantly increase the quality of the paper.
- The authors only analysed variants with single mutations. As shown in their Table S4, previously reported enhanced FAST variants displayed most of the time multiple mutations that can act in a synergistic manner. Have they tried to combine some of their mutations to further increase the gain in brightness ?
- The authors report on Fig 4 a comparison of the relative brightness for FAST mutants with different chromophores. The authors only show fluorescence micrographs of FAST/R52L (mcFAST-L) with MCL.1f and MCL.3a in SI on Figure S25. I would suggest to present representative micrographs of all experiments used for Fig 4, in order to better evaluate the performances of the different systems.
- The authors focused their study on improving the brightness in vitro and in cells. Brightness is however not the only parameter important for a fluorescent probe. It would strengthen the paper to include data comparing the new systems and the original one in terms of photostability and eventually phototoxicity, as the introduced mutations could affect both parameters.
- The authors mention in the discussion that several FAST variants exist now, such as the recently described pFAST that show high chromophore promiscuity and superior properties (Benaissa et al. Nat Commun 2021). It would be interesting to know if pFAST can bind N871b and its analogs and how does it compares to mcFAST-Y and mcFAST-L.

Minor remark

- Table S4 should reference the papers describing the listed variants, and the color code should be explained.

Reviewer #2 (Remarks to the Author):

In this manuscript, Goncharuk et al. describe a structure-based rational design approach for deriving mutants of a fluorogen-activating protein FAST for its N871b fluorogen. Using NMR and MD simulations, they define flexible hotspots within FAST which were serially mutated to finally obtain 2 point mutants with enhanced properties. They show that the mutants show enhancement for N871b as well as other related fluorogens. Finally, a rationale for the improved performance of

the mutants is provided using 1D-NMR experiments and MD simulations.

Generally, the data are interesting and the improved FAST mutants would be useful for in vivo studies. The data and analysis appear sound, however, I have pointed out some things that make the manuscript in its present form a little difficult to understand and needs improvement for better readability.

1. Figure 1 has a typo- us-ms timescale instead of ms-ms. Residues 52-53 are not marked in the top panel of the figure.

The Rex has been plotted on the structure but it would be good to provide a residue wise plot in the supplement.

The cut-off for flexibility should also be marked in the Supplementary figure S1. As it is presented now, one would assume that regions between H4-H5 or b3-b4 are also quite flexible.

2. Lines 114-115: these mutants which were chosen as they were favorable for other fluorogens should be plotted on the structure of FAST-N871b complex to understand the general proximity of these residues to the binding site.

3. According to the authors, only two mutants R52Y and R52L provide an improvement in extinction and quantum yield. However, R52A also has an improvement in both parameters even though it has a lower binding constant - can this be rationalized somehow? The NMR derived H-bond distance and stability as an indicator of better optical performance also does not fit for this mutant. The authors should address this discrepancy.

4. Supplementary figure S25 shows live-cell experiments pertaining to FAST R52L mutant. The authors should provide these data for R52Y mutant as well. Also Figure 4 contains box plot for MCL.1b fluorogen as well- raw data should also be provided for this.

5. Lines 191-193: these intermolecular hydrogen bonds could be shown in a figure, otherwise it is hard to follow where these residues might be located and the reader might be lost.

6. Figure 5- it is unclear why P68T is shown in the figure from the text currently- probably because it has the worst optical properties which agrees with NMR data as well. This should be mentioned in the text or the figure legend.

7. The authors conclude that intermolecular H-bond length with W94 side chain and stability of H-bonds formed by E46 and Y42 are indicators of better optical properties of FAST mutants-N871b complex. Can a complementary technique be used to show that the overall thermal stability of the complex is also improved in the brightest mutants? For eg. with NanoDSF or thermal shift assay.

8. A correlation of W94 chemical shift with brightness of the FAST mutant-N871b complex of 0.57 is still weak in my opinion and the stability of E46, and Y42 H-bonds even weaker. I would therefore modify the text to make this reasoning suggestive only for the brightest complexes. In the current form, it reads as if a metric for all the mutants tested.

9. In general, figure legends should be more descriptive and complete and are unacceptable in the current form. Good examples are- Table S4- what does the color code represent? Supplementary figure S1 mentions yFAST instead of FAST- this is probably a typo? What does the color code in Figure 5b,c,d represent?

10. Lines 99-101: would be good to include a supplementary figure for a comparison of the chemical structure of N871b with its parent fluorogens.

11. Supplementary figures S5-S10 could be combined. Also supplementary figures are not referenced in the text serially- for example S21 in line 126 is referenced right after S10. This creates problems with readability of the manuscript.

Reviewer #3 (Remarks to the Author):

This work done by Goncharuk et al. utilizes NMR relaxation experiments and MD simulation to identify "hot spots (dynamic region)" of FAST which helps improve the efficiency of enhanced mutation (brightness enhancement) selection. The authors also discussed the advantage and limitation of this approach which provides useful information for future technique development. However, I would like the authors to address/clarify followings in the manuscript which will be benefit for the field.

1. Protein production – In this study, 21 mutants of FAST were generated and characterized. The authors showed gel images of the proteins in the supplementary information. However, the mutations are all single site mutation, it's hard to know the success of mutation only by gel. Any mass spec data to support that the mutant variants of FAST are matched to their theoretical molecular weight?
2. The H5 residues are also in the mobile region and relative near to the N871b binding site, why not chose them as the mutation of interest? (Figure 7 also shows that by other publication, this region has enhanced FAST variants)
3. The rationalization of choosing residue 65 as a target is not clear. It's not in the fast sidechain motions regions identified by NMR or MD. The E46 and W94 of FAST are direct interact with N871b, why not chose them as mutation of interest to enhance the binding or fluorescence?
4. The NMR relaxation experiments are all done under condition - pH7 20mM NaPi + 20mM NaCl which is different from fluorescence or dissociation constants measurement buffer (PBS pH7.4 150mM NaCl). The loop regions' dynamic(mobility) in NMR are very sensitive to environment (pH and salt concentration etc.). To better guidance the selection of mutation site, the experiments buffer condition should be matched.

Minor:

1. In Figure 1, label residue 52, 46 and 94 will be easier for the audience to understand the idea.

Response to the reviewers

First of all, we would like to thank the editor and reviewers for their expertise and opinion. We have received very constructive comments that, we hope, will improve our manuscript. To answer the comments of the reviewers, we modified substantially the text and figures of the main text and supplementary data, performed several additional experiments. In particular, we measured the ^{15}N -relaxation for mcFAST-Y in complex with **N871b**, investigated the photostability of the proposed mutants, measured the brightness of pFAST in complex with our ligands, measured the brightness of all the bright mutants in NMR buffer, and studied the thermostability of FAST and its mutants with nanoDSF. Below we provide a point-by-point response to all the comments. In order to track the changes we uploaded a version of the manuscript, where all the changes that were introduced are marked by cyan.

Reviewer #1.

1. Although I agree that solving the NMR structure and measuring the backbone mobility of all 21 FAST variants analysed in this study was probably not necessary, it would have been very useful to do this analysis at least for the best variant mcFAST-Y (or mcFAST-L) in order to demonstrate the effect of the mutations and support their rational approach. Addition of such analysis would significantly increase the quality of the paper.

- Indeed, solving the structure of the brighter complex could justify our approach. However, the accuracy of NMR spectroscopy would not allow detecting the slightest changes in the lengths of hydrogen bonds that are expected to occur, according to the observed differences in chemical shifts. Thus, we suggest that resolving the NMR structure is not feasible. However, we can measure the ^{15}N relaxation of the mutant FAST in complex with **N871b** and investigate what happens with the mobility of the complex. To take into account this comment of the reviewer, we synthesized the ^{15}N -labeled mcFAST-Y, performed the backbone assignment and measured the NMR relaxation. According to the chemical shifts, the structure of the mutant did not change substantially, drastic chemical shift perturbations are observed only for the residues in direct proximity to R52. As revealed by NMR relaxation, the overall pattern of intramolecular mobility is the same for mcFAST-Y and FAST. However, the Rex values, which are indicative of slow μs -ms motions became significantly smaller both in the N-terminus and in the helix H5 of mcFAST-Y, and NH order parameters reveal the stabilization of Y52-D53 residues. The obtained data are described in the Results section of the manuscript (lines 274-283) and are shown in the revised figure 6 and supplementary figures S21-S22.

2. The authors only analysed variants with single mutations. As shown in their Table S4, previously reported enhanced FAST variants displayed most of the time multiple mutations that can act in a synergistic manner. Have they tried to combine some of their mutations to further increase the gain in brightness?

- Indeed, we planned to combine the mutations, if substitutions at various sites were favorable. However, as we observed, only the mutations at R52 appeared favourable, which does not provide any rational basis for a combination of mutations.

3. The authors report on Fig 4 a comparison of the relative brightness for FAST mutants with different chromophores. The authors only show fluorescence micrographs of FAST/R52L (mcFAST-L) with MCL.1f and MCL.3a in SI on Figure S25. I would suggest to present representative micrographs of all experiments used for Fig 4, in order to better evaluate the performances of the different systems.

- We provided the micrographs in Figure S17 of the revised manuscript.

Figure S17. Representative images of HeLa Kyoto cells transiently transfected with H2B-TagBFP-FAST (upper row), H2B-TagBFP-mcFAST-L (middle row), and H2B-TagBFP-mcFAST-Y (lower row) used to evaluate relative brightness of FAST mutant

variants with different chromophores in mCherry channel (N871b, MCL.1b, MCL.1f) and CY5 channel (MCL.3a); Concentration was 20 μ M for each chromophore; Scale bar is 20 μ m.

4. The authors focused their study on improving the brightness *in vitro* and in cells. Brightness is however not the only parameter important for a fluorescent probe. It would strengthen the paper to include data comparing the new systems and the original one in terms of photostability and eventually phototoxicity, as the introduced mutations could affect both parameters.

- We measured the photostability of FAST, mcFAST-Y and mcFAST-L in living cells, and almost no difference between the proteins taken in complexes with **N871b** was observed. The results are provided in the text (lines 169-171) of the revised manuscript and are shown in the new Figure S18.

Figure S18. Comparison of the photobleaching curves of FAST variants in the presence of 20 μ M of N871b. Solid lines represent mean value, shaded - standard deviation, n = 27 nuclei for each mutant variant.

5. The authors mention in the discussion that several FAST variants exist now, such as the recently described pFAST that show high chromophore promiscuity and superior properties (Benaissa et al. Nat Commun 2021). It would be interesting to know if pFAST can bind N871b and its analogs and how does it compares to mcFAST-Y and mcFAST-L.

- To answer this comment of the reviewer, we synthesized the pFAST protein, purified it and measured the brightness of pFAST in complexes with four ligands, used in our work: **N871b**, **MCL.1b**, **MCL.1f** and **MCL.3a**. In all the cases, pFAST demonstrated a significantly higher binding affinity than our proteins. However, the brightness of pFAST was comparable but not superior to the mutants, proposed in the present

work. In the case of **MCL.3a**, pFAST did not provide a substantial brightness enhancement, while R52L mutant did. Thus, our rational approach allowed constructing the FAST mutant with the brightness, comparable or superior to pFAST, found by the directed evolution as a protein, optimized for a wide variety of ligands. We included these data to the results section (lines 192-203) and to the discussion of the revised manuscript (lines 345-348). Now, brightness of pFAST complexes is provided in Table 1, and other data on the pFAST fluorescence are shown in supplementary Figures S8-S15.

Minor remark

6. *Table S4 should reference the papers describing the listed variants, and the color code should be explained.*

- We added the reference and included the color code description to the legend of Table S4 (S5 in the revised version).

Reviewer #2.

1. *Figure 1 has a typo- us-ms timescale instead of ms-ms. Residues 52-53 are not marked in the top panel of the figure.*

The Rex has been plotted on the structure but it would be good to provide a residue wise plot in the supplement.

The cut-off for flexibility should also be marked in the Supplementary figure S1. As it is presented now, one would assume that regions between H4-H5 or b3-b4 are also quite flexible.

- We marked R52 in the top panel of Figure 1 and corrected the typo. Rex values were added to Figure S1, as well as the S^2 cut-offs for the flexibility. Regions between H4-H5 (70-73) and B3-B4 (98-102) are indeed quite flexible, this is actually stated in the text, and several of our mutations were made in an attempt to stabilize these regions. *"Analysis of the relaxation parameters with a model-free approach reveals three major mobile regions in close proximity to the ligand - loops 98-102, 70-73, and 52-53 (Fig. 1)."*

Revised figure 1. Dynamics of FAST/N871b complex.

2. Lines 114-115: these mutants which were chosen as they were favorable for other fluorogens should be plotted on the structure of FAST-N871b complex to understand the general proximity of these residues to the binding site.

- To take into account this comment of the reviewer, we made an additional panel for Figure 2 (Figure 2B in the revised version):

Revised figure 2. Mutagenesis of FAST.

3. According to the authors, only two mutants R52Y and R52L provide an improvement in extinction and quantum yield. However, R52A also has an improvement in both parameters even though it has a lower binding constant - can this be rationalized somehow? The NMR derived H-bond distance and stability as an indicator of better optical performance also does not fit for this mutant. The authors should address this discrepancy.

- Indeed, R52A appeared brighter than FAST in complex with **N871b**, while the intermolecular hydrogen bonds are unstable in this complex, according to the NMR analysis. However, we need to note that our approach relies on the correlations and the correlation coefficients are not close to 1, implying that there are outliers. For instance, there is a R52F mutant, that should be brighter than FAST, judging by NMR data, but it is not, and vice versa, R52A, which should be dimmer, but it is brighter.

This implies that there are definitely other factors besides those found in the work that affect brightness that we don't know about. We have changed the results section to indicate this explicitly (lines 228-232).

4. Supplementary figure S25 shows live-cell experiments pertaining to FAST R52L mutant. The authors should provide these data for R52Y mutant as well. Also Figure 4 contains box plot for MCL.1b fluorogen as well- raw data should also be provided for this.

- We added the raw data for all the live-cell experiments to the supplementary Figure S17 (shown above).

5. Lines 191-193: these intermolecular hydrogen bonds could be shown in a figure, otherwise it is hard to follow where these residues might be located and the reader might be lost.

- We added the new panel f to the figure 5, which displays all three key hydrogen bonds.

Revised figure 5. NMR analysis of FAST/N871b complexes.

6. Figure 5- it is unclear why P68T is shown in the figure from the text currently- probably because it has the worst optical properties which agrees with NMR data as well. This should be mentioned in the text or the figure legend.

- Yes, P68T is provided as an example of the worst mutant. We now mention it in the figure legend: "Overlay of *1H* NMR spectra of FAST/N871b complexes, recorded for the wild-type protein and its three mutants: R52Y, R52L (the best mutants) and P68T (the worst mutant)."

7. The authors conclude that intermolecular H-bond length with W94 side chain and stability of H-bonds formed by E46 and Y42 are indicators of better optical properties of FAST mutants-N871b complex. Can a complementary technique be used to show that the overall thermal stability of the complex is also improved in the brightest mutants? For eg. with NanoDSF or thermal shift assay.

- First of all, we would like to point out, that the stability we are seeking for - is stability in terms of intramolecular motions and it may differ from both the stability of the ground state and the thermal stability of the complex. The overall stability of the latter can be also assessed from the dissociation constants, and it has evidently increased for the brighter mutants. However, the overall conformational stability of the complex (Kd) does not correlate with the brightness, if we take all the mutants into account. Following the suggestion of the reviewer, we have measured the thermal stability of FAST, mcFAST-Y and mcFAST-L, both in the apo-state and in complex with **N871b** by nanoDSF using Prometheus Panta in nanoDSF grade standard capillaries (NanoTemper Technologies GmbH, München, Germany). As we found out, the melting of proteins in the presence and in the absence of **N871b** occurs at the same temperature, suggesting that complexes dissociate at lower temperatures, probably due to the temperature dependence of the dissociation constant and limited solubility of the fluorogen in water. In other words, we are able to detect only the melting of the protein apo state, and the obtained data reveal that this ligand-free form is stabilized by both the R52Y and R52L mutations. While R52Y increases the melting temperature by $1.4 \pm 0.2^\circ$, R52L provides the stabilization by $3.5 \pm 0.2^\circ$. We included these results to the revised version (Figure S23, lines 282-290), and this required the addition of two authors - Valentin Borshchevskiy and Alexey Mishin.

Figure S23. Representative curves for the normalized derivative of fluorescence emission ratio at 350/330 nm as a function of temperature. Curves for FAST shown in black, mcFAST-L in blue, mcFAST-Y in green, apo forms shown as solid lines, **N871b** bound forms - as dashed lines. The table shows T_m (°C) values for FAST, mcFAST-L and mcFAST-Y both in apo and **N871b** bound states. Data represent the mean \pm SD (n=3).

8. A correlation of W94 chemical shift with brightness of the FAST mutant-N871b complex of 0.57 is still weak in my opinion and the stability of E46, and Y42 H-bonds even weaker. I would therefore modify the text to make this reasoning suggestive only for the brightest complexes. In the current form, it reads as if a metric for all the mutants tested.

- We added the sentence to the results section which states that the correlations are weak and that there are other factors. Now we state (lines 228-232): "*To summarize, the brightest mutants correspond to the complexes with the shortest intermolecular H-bonds formed by W94 side chains and the highest stability of intermolecular H-bonds formed by E46 and Y42. However, there are outlying mutants - R52A, which is brighter than FAST, but does not reveal any NMR properties of brightest mutants, and R52F, which demonstrates the characteristics of the bright mutant, but is dimmer than the original FAST tag. This, together with the rather weak character of*

correlations, suggest that there are other factors that influence the brightness of FAST/N871b complex that we still do not understand."

9. In general, figure legends should be more descriptive and complete and are unacceptable in the current form. Good examples are- Table S4- what does the color code represent? Supplementary figure S1 mentions *y*FAST instead of FAST- this is probably a typo? What does the color code in Figure 5b,c,d represent?

- we checked all the figure legends, and corrected all the flaws, found by the reviewer.

10. Lines 99-101: would be good to include a supplementary figure for a comparison of the chemical structure of N871b with its parent fluorogens.

- We added a figure with chemical structures of conventional rhodanine-based FAST ligands and GFP chromophores-like FAST fluorogens to the supplementary materials as Figure S2.

Figure S2. Chemical structures of rhodanine-based conventional FAST fluorogens, and GFP chromophore-based fluorogens, studied here.

11. Supplementary figures S5-S10 could be combined. Also supplementary figures are not referenced in the text serially- for example S21 in line 126 is referenced right after S10. This creates problems with readability of the manuscript.

- We combined the figures S5-S10 (they are Figure S6 in the revised version). We have also renumbered and rearranged all the supplementary figures to ensure that the numbering order corresponds to the occurrence of figure references in the main text.

Reviewer #3.

1. Protein production – In this study, 21 mutants of FAST were generated and characterized. The authors showed gel images of the proteins in the supplementary information. However, the mutations are all single site mutation, it's hard to know the success of mutation only by gel. Any mass spec data to support that the mutant variants of FAST are matched to their theoretical molecular weight?

- The main support in favor of the correct mutations is DNA sequencing. All our mutant constructs were confirmed by DNA sequencing. We added this clarification to the text (line 379). Several independent sample preparations (including transformation, cultivation, lysis, purification etc.) eliminate the possibility of accidental error like mixing tube samples, because in all cases each protein behavior was similar.

2. The H5 residues are also in the mobile region and relative near to the N871b binding site, why not chose them as the mutation of interest? (Figure 7 also shows that by other publication, this region has enhanced FAST variants)

- Indeed, H5 is mobile, but H5 residues are not in direct contact with the ligand. As we have stated in the text, we selected those residues that are located in the mobile regions and are in direct contact with the pyridine group of the ligand. There are many options for mutagenesis - one can try to stabilize the H5 helix, to stabilize the N-terminus or to affect directly the ligand binding by mutating the residues in the hydrophobic core of the protein. However, this provides multiple variants that need to be tested and such an approach would be closer to the random mutagenesis than to the rational design. Since all the options were tested previously in directed evolution for other ligands, we decided to focus on the options that are peculiar for **N871b**, and test only the mutants that could favorably interact with this moiety.

3. The rationalization of choosing residue 65 as a target is not clear. It's not in the fast sidechain motions regions identified by NMR or MD. The E46 and W94 of FAST are direct interact with N871b, why not chose them as mutation of interest to enhance the binding or fluorescence?

- In part, we have already answered this comment above. We focused on the pyridine group of **N871b**, while E46 and W94 are the key residues that interact with other parts of the ligand. These parts - substituted phenolic ring and imidazolone ring are quite similar among all the FAST ligands, and we assumed that if the improvements at these positions of FAST are possible, they should have been detected in the course of directed evolution optimization with other ligands.

As for D65, we hypothesized that this residue could stabilize R52 via a salt bridge. Thus we inverted the charge at this position (D65R and D65K), and designed a

double mutant with both D65 and R52 charges being inverted (R52E/D65R). To clarify, we modified the Results section in the revised manuscript (lines 114-116).

4. The NMR relaxation experiments are all done under condition - pH7 20mM NaPi + 20mM NaCl which is different from fluorescence or dissociation constants measurement buffer (PBS pH7.4 150mM NaCl). The loop regions' dynamic(mobility) in NMR are very sensitive to environment (pH and salt concentration etc.). To better guidance the selection of mutation site, the experiments buffer condition should be matched.

- We agree with the reviewer, that we need to match the conditions. To do so, we measured the brightness of FAST/N871b and two best mutants in the NMR buffer. While the absolute values were different, the relationship between the proteins remained the same - both mcFAST-Y and mcFAST-L remained brighter than FAST, the brightness of mcFAST-Y and mcFAST-L were comparable. We modified the Results section of the manuscript to indicate this (lines 132-134), and show the results in supplementary materials Table S3 and Figure S5.

Table S3. Optical properties of N871b in complexes with FAST mutants measured in various buffers.

Buffer	Mutant	K_D , μM ^a	ϵ , $\text{M}^{-1}\cdot\text{cm}^{-1}$ ^b	FQY, % ^c	Brightness
PBS buffer, pH 7.4, #cat E404-200TABS, Amresco	FAST	0.33±0.01	27000±410	26±1.4	7000±480
	R52L	0.27±0.01	30000±450	29±1.7	8750±640
	R52Y	0.24±0.02	29000±440	29±1.5	8350±560
pH 7.0, 20 mM NaPi, 20 mM NaCl	FAST	0.21±0.01	26000±390	29±1.1	7500±400
	R52L	0.19±0.02	28500±430	31±1.2	9000±480
	R52Y	0.17±0.03	27500±410	34±2.3	9400±780

a – represented as mean ± SD (n = 3);

b – represented as result of single measurement ± the precision of the measuring instruments (weighing and pipetting errors);

c – fluorescence quantum yield, represented as mean ± SD (n = 9).

Minor:

1. In Figure 1, label residue 52, 46 and 94 will be easier for the audience to understand the idea.

- We labeled R52 in figure 1 of the revised version, as requested. E46 and W94 are not mentioned before Figure 5, therefore we added a panel to Figure 5, which shows all three essential residues that take part in the intermolecular hydrogen bonding.

REVIEWERS' COMMENTS:

Reviewer #2 (Remarks to the Author):

Goncharuk et al. addressed all the issues I had with the manuscript previously. I however found 1 minor point- Figure 5 where the new panel f is added is neither referenced in the text nor mentioned in the figure legend in the current version.

Reviewer #3 (Remarks to the Author):

The authors have addressed all my questions and concerns. I'd like to recommend this work to be published in Communications Biology.

First of all, we would like to thank the reviewer for their comments and positive decision. Below is the answer to the only left comment of the reviewer #2

- Figure 5 where the new panel f is added is neither referenced in the text nor mentioned in the figure legend in the current version.

- We modified the Figure 5 (Figure 4 in the revised version). Now the unreferenced panel is indicated by letter "a", is described in the caption and referenced in the text of the manuscript.